# META-LEARNING WITH DOMAIN ADAPTATION FOR FEW-SHOT LEARNING UNDER DOMAIN SHIFT

## ABSTRACT

Few-Shot Learning (learning with limited labeled data) aims to overcome the limitations of traditional machine learning approaches which require thousands of labeled examples to train an effective model. Considered as a hallmark of human intelligence, the community has recently witnessed several contributions on this topic, in particular through meta-learning, where a model learns how to learn an effective model for few-shot learning. The main idea is to acquire prior knowledge from a set of training tasks, which is then used to perform (few-shot) test tasks. Most existing work assumes that both training and test tasks are drawn from the same distribution, and a large amount of labeled data is available in the training tasks. This is a very strong assumption which restricts the usage of meta-learning strategies in the real world where ample training tasks following the same distribution as test tasks may not be available. In this paper, we propose a novel meta-learning paradigm wherein a few-shot learning model is learnt, which simultaneously overcomes domain shift between the train and test tasks via adversarial domain adaptation. We demonstrate the efficacy the proposed method through extensive experiments.

## 1 INTRODUCTION

Few-Shot Learning aims to learn a prediction model from very limited amount of labelled data (Lake et al., 2015). Specifically, given a $K-$shot, $N-$class data for a classification task, the aim is to learn a multi-class classification model for $N-$ classes, with $K-$labeled training examples for each class. Here $K$ is usually a small number (e.g. 1, or 5). Considered as one of the hallmarks of human intelligence (Lake et al., 2011), this topic has received considerable interest in recent years (Lake et al., 2015; Koch et al., 2015; Vinyals et al., 2016; Finn et al., 2017). Modern techniques solve this problem through *meta-learning*, using an *episodic* learning paradigm. The main idea is to use a labeled training dataset to effectively acquire prior knowledge, such that this knowledge can be transferred to novel tasks where few-shot learning is to be performed. Different from traditional transfer learning (Pan et al., 2010; Yosinski et al., 2014), here few-shot tasks are simulated using the labeled training data through episodes, in order to acquire prior knowledge that is specifically tailored for performing few-shot tasks. For example, given a set of labeled training data with a finite label space $\mathcal{Y}^{train}$, the epsiodic paradigm is used to acquire prior knowledge which is stored in a model. Each episode is generated i.i.d from an unknown task distribution $\tau_{train}$. This model is then used to do a novel few shot classification task which is drawn from an unknown task distribution $\tau_{test}$. The test task comprises small amount of labeled data with a finite label space $\mathcal{Y}^{test}$, and the sets $\mathcal{Y}^{train}$ and $\mathcal{Y}^{test}$ are (possibly) mutually exclusive. Using this labeled data, and acquired prior knowledge, the goal is to predict the labels of all unlabeled instances in the test task.

A very restrictive assumption of existing meta-learning approaches for few-shot learning is that train and test tasks are drawn from the same distribution, i.e., $\tau_{train} = \tau_{test}$. In this scenario, the meta-learner's objective is to minimize its expected loss over the tasks drawn from the task distribution $\tau_{train}$. This assumption prohibits the use of meta-learning strategies for real-world applications, where training tasks with ample labeled data, and drawn from the same distribution as the test tasks are very unlikely to be available. Consider the case of a researcher or practitioner who wishes to train a prediction model for their own dataset where labeled data is very limited. It is unreasonable to assume that they would have a large corpus of labeled data for a set of related tasks in the same domain. Without this, they are not able to train effective few-shot models for their task. A more desirable option is to use the training tasks where ample training data is available, and adapt the

model to be effective on test tasks in a different domain. A possible way to tackle this problem could be through the use of domain adaptation techniques (Ganin et al., 2016; Hoffman et al., 2018) that address the domain shift between the training and test data. However, all of these approaches address the single-task scenario, i.e., $\mathcal{Y}^{train} = \mathcal{Y}^{test}$, where the training data and test data are sampled from the same task but there is a domain shift at a data-level. This is in contrast to the meta-learning setting where the training data contains multiple tasks and the goal is to learn new tasks from test data, i.e., domain shift exists at a task-level and $\mathcal{Y}^{train} \cap \mathcal{Y}^{test} = \emptyset$. As a result, these domain adaptation approaches cannot be directly applied. We show an overview of different problem settings in Table 1.

Table 1: Illustration of the differences between our work and the other three lines of work.

|  | Training | Test | Domain Shift |
|---|---|---|---|
| **Standard Supervised Learning** | Task 1 | Task 1 | no |
| **Domain Adaptation (DA)** | Task 1 | Task 1 | instance-level shift |
| **Meta Learning** | Task 1$\cdots$ Task $N$ | Task $N+1\cdots$ | no |
| **Meta Learning with DA** | Task 1$\cdots$ Task $N$ | Task $N+1\cdots$ | task-level shift |

In order to solve the few-shot learning problem under a domain shift we propose a novel meta-learning paradigm: *Meta-Learning with Domain Adaptation (MLDA)*. Existing meta-learning approaches for few-shot learning use only the given training data to learn a model, and as a result they do not account for any domain shift between the training tasks and the few-shot test tasks. In contrast, we assume that the model has access to the unlabeled instances in the domain of the few-shot test tasks prior to the training procedure, and utilize these instances for incorporating the domain-shift information. We train the model under the episodic-learning paradigm, but in each episode we aim to train a model which achieves two goals: first the model should be good at few-shot learning, and second the model should be unaffected by a possible domain shift. The first goal is achieved by updating the model based on the few-shot learning loss suffered by the model for a given episode. The second goal is achieved by an adversarial domain adaptation approach, where a mapping is used which *styles* the training task to resemble the test task. In this way, the trained model can perform few-shot predictions on the test tasks, and achieve what we term *task-level* domain adaptation.

The episodic update is done via Prototypical Networks (Snell et al., 2017) (as a specific instantiation, though other approaches can be applied), where on a simulated few-shot task (a small support set behaves as training, and a query set behaves as test data), an embedding is produced for both support and query instances. The mean of support embedding of each class is the prototype, and query instances are labeled based on their distance to these prototypes. Based on the loss on these query instances, the embedding function is updated. For achieving invariance to domain shift, we follow the principle of adversarial domain adaptation, but we differ from the traditional approaches in that we are performing task-level domain adaptation, whereas they performed data-level domain adaptation. The early approaches to adversarial domain adaptation aimed at obtaining a feature embedding that was invariant to both the training domain and the test domain, as well as learning a prediction model in the training domain (Ganin et al., 2016). However, these approaches possibly learnt a highly unconstrained feature embedding (particularly when the embedding was very high dimensional), and were outperformed by GAN-based approaches (often used for image translation) (Taigman et al., 2017; Zhu et al., 2017; Hoffman et al., 2018). As a result we use a mapping function to style the training tasks to resemble test tasks, and optimize it using a GAN loss. The overall framework delivers a model that uses training tasks from one distribution to meta-learn a few-shot model for a task from another distribution. We perform extensive experiments to show the efficacy of the proposed method.

## 2 RELATED WORK

### 2.1 META-LEARNING FOR FEW-SHOT LEARNING

Few-Shot Learning refers to learning a prediction model from small amount of labeled data (Fei-Fei et al., 2006; Lake et al., 2011). Early approaches used a Bayesian model (Fei-Fei et al., 2006), or hand-designed priors (Lake et al., 2015). More recently, meta-learning approaches have become extremely successful for addressing few-shot learning (Vinyals et al., 2016; Finn et al., 2017). Instead of training a model directly on the few-shot data, meta-learning approaches use a corpus of labeled data, and simulate few-shot tasks on them to learn how to do few-shot learning. Some approaches follow the non-parametric principle, and develop a differentiable $K-$nearest neighbour solution (Vinyals et al., 2016; Shyam et al., 2017; Snell et al., 2017). The main concept is to learn an embedding space that is tailored for performing effective K-nearest neighbour. Oreshkin et al. (2018)

extend these approaches with metric scaling to condition the embedding based on the given task. Another category of meta-learning aims to learn how to quickly adapt a model in few gradient steps for a few-shot learning task (Finn et al., 2017; Ravi & Larochelle, 2017; Li et al., 2017). These optimization based approaches aim to learn an initialization from a set of training tasks, which can be quickly adapted (e.g. one-step gradient update) when presented with a novel few-shot task. Some other approaches consider using a "memory"-based approach (Santoro et al., 2016; Munkhdalai & Yu, 2017). There have also been approaches that try to enhance meta-learning performance through use of additional information. For example, Ren et al. (2018) use unlabeled data to develop semi supervised few-shot learning. Zhou et al. (2018) use external data to generate concepts, and performs meta-learning in the concept space. However, all of these approaches assume that the training tasks and testing tasks are drawn from the same distribution ($\tau_{train} = \tau_{test}$). If there is a task-level domain shift, the above approaches will fail to perform few-shot learning on novel test tasks. Our approach of meta-learning with domain adaptation overcomes this domain shift, to perform few-shot learning on tasks in a different domain.

## 2.2 DOMAIN ADAPTATION

Domain adaptation has been studied extensively in recent years, particularly for computer vision applications (Saenko et al., 2010). The idea is to exploit labeled data in one domain (called source domain) to perform a prediction task in another domain (called the target domain), which does not have any labels (unsupervised domain adaptation). Most approaches employed two objectives: one to learn a prediction model in the source domain, and second to find an embedding space between the two domains that achieves domain invariance, thus making the model trained on the source domain applicable to the task in the target domain. In the era of deep learning, some early approaches aimed to align feature distribution in some embedding space using statistical measures (e.g. Maximum Mean Discrepancy) (Tzeng et al., 2014; Long et al., 2015). This was followed by several successful efforts for domain adaptation using an adversarial loss (Goodfellow et al., 2014). Ganin & Lempitsky (2015); Ganin et al. (2016) aimed to learn a feature embedding such that a domain classifier would not be able to distinguish whether the instance was drawn from the source or target domain. Consequent efforts tried to learn an embedding on the source data, from which an instance in the target domain could be reconstructed (Ghifary et al., 2016). Tzeng et al. (2017) proposed to train a model in the source domain, and using a GAN loss try to embed the target domain to the same feature distribution (using a GAN loss) as the (now fixed) source domain. Another line of work using GAN-loss is for image-to-image translation, where images in one domain are mapped to appear like images in another domain (Taigman et al., 2017; Liu et al., 2017). Most of these approaches have demonstrated application to domain adaptation tasks as well. Another recently introduced concept is cycle consistency which first maps an instance from the source to target, and then maps this synthetic instance back to the source to get back to original instance (Zhu et al., 2017; Kim et al., 2017; Yi et al., 2017), and this concept has been extended for domain adaptation as well (Hoffman et al., 2018). All of these approaches aim to solve the same task in both domains (i.e., the label space is the same in both domains). They perform domain adaptation at the data-level (and not the task level). This means that they cannot solve a new task with a different label space. In contrast our approach performs a task-level domain adaptation, and can solve new tasks.

There have been some efforts at the intersection of few-shot (and meta-learning) and domain adaptation. Motiian et al. (2017) consider supervised domain adaptation, which is similar to unsupervised domain adaptation setting, except that few labeled instances in the target domain are available. Like the previous approaches, it can not be used for a novel task with a different label space. Kang & Feng (2018)'s problem setting resembles traditional unsupervised domain adaptation, except that the model training is done using a meta-learning principle. Li et al. (2018) use meta-learning to address domain generalization where a single trained model for a given task, can be applied to any new domain with a different data distribution. They too consider solving the same task in a new domain, and do not consider the few-shot setting. A closely related work is Domain Adaptive Meta Learning (Yu et al., 2018), but their problem setting is different (which is more suitable for the problem they address: imitation learning) from what we address in this paper. They consider the scenario where a task has training data drawn from one domain and test data drawn from another domain (independent of whether it has been drawn from $\tau_{train}$ or $\tau_{test}$). Thus, they do not violate $\tau_{train} = \tau_{test}$. In their simulated task, ample labeled training data is available for both the source and target domains. In contrast, we consider the scenario where the training tasks and test tasks are drawn from different distributions, and we have very limited labeled data for test tasks (tasks in the target domain).

## 3 META-LEARNING WITH DOMAIN ADAPTATION (MLDA)

Formally, let $\mathcal{X}$ be an input space and $\mathcal{Y}$ be a discrete label space. Let $\mathcal{D}$ be a distribution over $\mathcal{X} \times \mathcal{Y}$. During meta-training, the meta-learner has access to a large labeled dataset $S_{train}$ that typically contains thousands of instances for a large number of classes $C$. At the $i$-th iteration of *episodic* learning paradigm, the meta-learner samples a $K-$shot $N$-class classification task $T_i$ from $S_{train}$, which contains a small "training" (assumed to have labels for all instances for this task) set $S_i^{support}$ (with $K$ examples from each class) and a "test" (assumed to not have labels of any instances for this task) set $S_i^{query}$. Both $S_i^{support}$ and and $S_i^{query}$ are assumed to be generated from an unknown sample distribution $S_i^{support} \sim \mathcal{D}_i^m$ and $S_i^{query} \sim \mathcal{D}_i$ respectively, where $m = NK$ denotes the number of instances. The sample distribution $\mathcal{D}_i$ are assumed to be generated *i.i.d.* from an unknown task distribution $\tau_{train}$, i.e, $((\mathcal{D}^m, \mathcal{D}) \sim \tau_{train})$. It then computes conditional probabilities $p(y|\mathbf{x}, S_i^{support})$ for every point $(\mathbf{x}; y)$ in the test set $S_i^{query}$. Based on these predictions, meta-learner incurs a loss $L(p(y|\mathbf{x}, S_i^{support}), y)$ for each point in the current $S_i^{query}$. The meta-learner then back-propagates the gradient of the loss $\sum_{(x,y) \in S^{query}} L(p(y|\mathbf{x}, S^{support}), y)$ for updating the model. In the meta-testing phase, the resulting meta-learner is used to solve the novel $K-$shot $N$-class classification tasks, which are assumed to be generated *i.i.d.* from an unknown task distribution $\tau_{test}$. The labeled training set and unlabeled test examples are given to the classification algorithm and the algorithm outputs class probabilities.

Existing meta-learning approaches assume that both training and testing tasks are drawn from the same distribution, i.e., $\tau_{train} = \tau_{test}$. However, this may not be the case in several real-world scenarios (i.e., $\tau_{train} \neq \tau_{test}$ ). Consider the case of a researcher who wants to do few-shot classification on a newly collected image recognition dataset (task drawn from $\tau_{test}$). This researcher must now find a large amount of labeled data from which tasks can be drawn from the same task distribution ($\tau_{test}$), failing which the researcher does not have a clear approach to acquire the relevant prior knowledge. The alternative is for the researcher to find tasks drawn from a different distribution, where ample labeled data is available, and perform task-level domain adaptation in order to learn a few-shot model suitable for their own task. Thus, we make a distinction between the task drawn from $\tau_{train}$ and $\tau_{test}$, as $(\mathcal{D}_{train}^m, \mathcal{D}_{train}) \sim \tau_{train}$ and $(\mathcal{D}_{test}^m, \mathcal{D}_{test}) \sim \tau_{test}$. $(\mathcal{D}_{train}^m, \mathcal{D}_{train})$ and $(\mathcal{D}_{test}^m, \mathcal{D}_{test})$ may have tasks whose instances are drawn from different domains ($\mathbf{x}^{train} \in \mathcal{X}^{train}$ and $\mathbf{x}^{test} \in \mathcal{X}^{test}$ respectively), and may also have a mutually exclusive discrete label space (e.g. $\mathcal{Y}^{train} \cap \mathcal{Y}^{test} = \emptyset$). Our overall goal is to learn a meta-learner that can utilize tasks drawn from $\tau_{train}$ to acquire a good prior for few-shot learning, and overcome the task-level domain-shift in order to learn unobserved few-shot tasks drawn from $\tau^{test}$. The general setting can be seen in Figure 1. Next, we briefly describe our proposed few-shot learning approach under task-level domain shift.

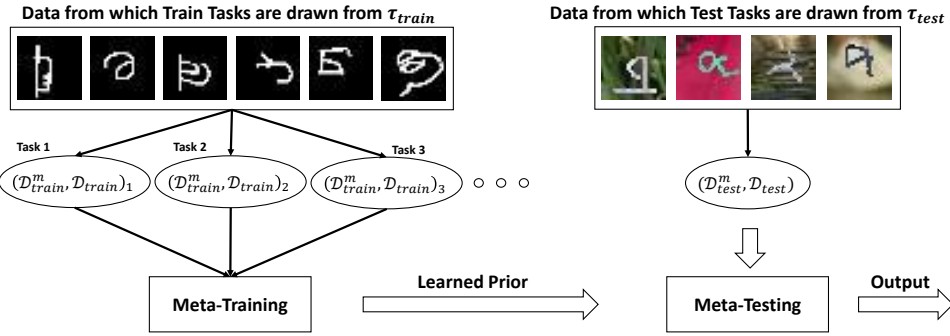

Figure 1: Problem Setting for Meta-Learning with Domain Adaptation. Tasks are drawn from $\tau_{train}$, on which meta learning is performed, such that the learner can do effective meta-testing for tasks drawn from a different distribution $\tau_{test}$. The images are adapted from the Omniglot dataset (Lake et al., 2011), where the left block has some original instances of hand-written characters in the original domain, and in the right block, we have a set of different omniglot characters (or classes) and the data is also in a different domain.

### 3.1 FEW-SHOT LEARNING IN NEW DOMAINS USING MLDA

Here, we give the overview of our proposed learning paradigm: *Meta Learning with Domain Adaptation (MLDA)*. We have two objectives that need to be optimized simultaneously. First, we want to learn a feature extractor that can learn discriminative features which can be used for few-shot learning on novel tasks. Second, we want these features to be invariant to the train task distribution and test task distribution, i.e., for a task $T_i \sim (\mathcal{D}_{train}^m, \mathcal{D}_{train})$, we want to adapt it to resemble a task drawn from $(\mathcal{D}_{test}^m, \mathcal{D}_{test})$. Specifically, in the meta-learning phase, we consider a feature extractor $\mathbf{F} : \mathcal{X}^{train} \to \mathbb{R}^d$ which takes an input instance $\mathbf{x} \in \mathcal{X}^{train}$ and returns a $d-$dimensional embedding. This feature extractor in turn is a composition function $\mathbf{F}(\mathbf{x}) = \hat{\mathbf{F}}(\mathbf{G}(\mathbf{x}))$, where $\mathbf{G} : \mathcal{X}^{train} \to \mathcal{X}^{test}$, and $\hat{\mathbf{F}} : \mathcal{X}^{test} \to \mathbb{R}^d$. The feature extractor $\mathbf{F}$ is trained to learn a representation suitable for few-shot learning (by optimizing objective $\mathcal{L}_{fs}$). $\mathbf{G}$ aims to achieve task-level domain invariance by translating instances from domain $\mathcal{X}^{train}$ to instances in domain $\mathcal{X}^{test}$. $\mathbf{G}$ is trained using an adversarial loss, inspired by recent success of GAN-based (Goodfellow et al., 2014) domain adapation methods (Tzeng et al., 2017; Zhu et al., 2017; Hoffman et al., 2018). $\mathbf{G}$ (along with the corresponding discriminator $D$) is trained to achieve domain adaptation (by optimizing objective $\mathcal{L}_{da}$). We also use a mapping $\mathbf{G}' : \mathcal{X}^{test} \to \mathcal{X}^{train}$ to obtain cyclic consistency, wherein we try to translate generated instance $\mathbf{G}(\mathbf{x})$ to produce the original instance $\mathbf{x}$. The overall objective function is given by:

$$\min_{\hat{\mathbf{F}},\mathbf{G},\mathbf{G}'} \max_{D} \mathcal{L}_{fs} + \mathcal{L}_{da} \tag{1}$$

Note that $\mathcal{L}_{fs}$ is optimized using only labeled training data of tasks drawn from $\tau_{train}$ and $\mathcal{L}_{da}$ is optimized using unlabeled data of tasks drawn from both $\tau_{train}$ and $\tau_{test}$. The overall framework can be seen in Figure 2. Next, we will describe motivation and technical details of these components.

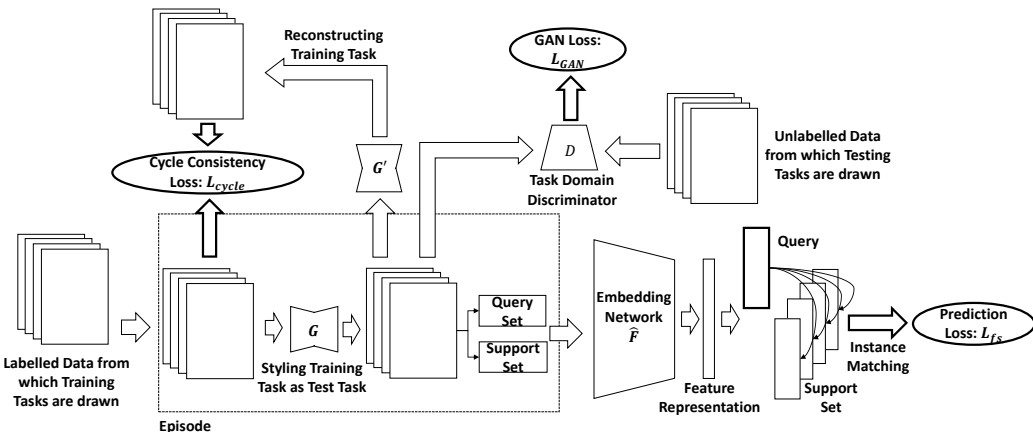

Figure 2: Meta Learning with Domain Adaptation (MLDA): The proposed method for few-shot learning under task-level domain shift using adversarial domain adaptation. A task sampled from $\tau_{train}$ in every episode. This task is used to update the parameters with the aim of achieving 2 goals: 1) It follows a Prototypical Networks learning scheme to acquire few-shot learning ability, and 2) It styles the task to appear indistinguishable from a task drawn from $\tau_{test}$. Task-level domain invariance is achieved through the usage of a GAN loss and a cycle-consistency loss ($\mathcal{L}_{da} = \mathcal{L}_{GAN} + \mathcal{L}_{cycle}$).

### 3.2 FEW-SHOT LEARNING

There have been several approaches for few-shot learning via meta-learning in literature (Vinyals et al., 2016; Finn et al., 2017; Snell et al., 2017). In principle, our proposed paradigm is agnostic to any of these approaches. In our work, we follow a recent state of the art approach: Prototypical Networks (Snell et al., 2017) to instantiate our framework for meta learning with domain adaptation. For a given task $T_i \sim (\mathcal{D}_{train}^m, \mathcal{D}_{train})$, Prototypical Networks use a feature extractor $\mathbf{F} : \mathcal{X}^{train} \to \mathbb{R}^d$ to

compute a $d-$dimensional embedding for each instance. Using this feature extractor, the mean vector embedding is computed for each class $\mathbf{c}_n$ for $n = 1, \ldots, N$, which are the prototypes of each class:

$$\mathbf{c}_n = \frac{1}{S_n^{support}} \sum_{(\mathbf{x}_i, y_i) \in S_n^{support}} \mathbf{F}(\mathbf{x}_i) \tag{2}$$

For a given query instance $\mathbf{x}$, Prototypical Network will produce a probability distribution over the classes using:

$$p(y = n | \mathbf{x}) = \frac{\exp(-dist(\mathbf{F}(\mathbf{x}), \mathbf{c}_n))}{\sum_{(j=1)}^{N} \exp(-dist(\mathbf{F}(\mathbf{x}), \mathbf{c}_j))} \tag{3}$$

where $dist : \mathbb{R}^d \times \mathbb{R}^d \to [0, \infty)$ is a function measuring the distance between the embeddings of a query instance and a class prototype. The few-shot loss $\mathcal{L}_{fs}$ is the negative log-probability:

$$\mathcal{L}_{fs} = -\log p(y = k | \mathbf{x}) \tag{4}$$

This loss is evaluated on the query set $S_i^{query}$, and backpropagated to update the parameters of feature extractor $\mathbf{F}$. In this setup, $\mathbf{F}$ does not account for a domain shift between $\tau_{train}$ and $\tau_{test}$. Consequently, we use $\mathbf{F}(\mathbf{x}) = \hat{\mathbf{F}}(\mathbf{G}(\mathbf{x}))$, where $\mathbf{G}$ will help incorporate the domain shift information.

### 3.3 ADVERSARIAL TASK-LEVEL DOMAIN ADAPTATION

Here, we describe how to perform task-level domain adaptation and learn the mapping parameters $\mathbf{G}$.

#### 3.3.1 GAN LOSS AND CYCLE CONSISTENCY

We use the GAN loss (Goodfellow et al., 2014) to learn the mapping $\mathbf{G} : \mathcal{X}^{train} \to \mathcal{X}^{test}$, and its corresponding discriminator $D$. The objective function is denoted as:

$$\mathcal{L}_{GAN}(\mathbf{G}, D, \mathcal{X}^{train}, \mathcal{X}^{test}) = \mathbb{E}_{\mathbf{x}^{test} \sim \mathcal{D}_{test}}[\log D(\mathbf{x}^{test})] + \tag{5}$$
$$\mathbb{E}_{\mathbf{x}^{train} \sim \mathcal{D}_{train}}[\log(1 - D(\mathbf{G}(\mathbf{x}^{train})))]$$

Here $\mathbf{G}$ tries to generate instances that appear to be similar to the instances in domain $\mathcal{X}^{test}$, and $D$ tries to distinguish between translated instances $\mathbf{G}(\mathbf{x}^{train})$ and the real samples $\mathbf{x}^{test}$. This objective is minimized under the parameters of $\mathbf{G}$ and maximized under the parameters of $D$. This effectively leads to translating tasks drawn from $\tau_{train}$ to be translated such that they are indistinguishable from tasks drawn from $\tau_{test}$.

Despite the ability of adversarial networks to produce an output indistinguishable from the test domain $\mathcal{X}^{test}$, with a large capacity, it is not inconceivable for the network to map the same set of input images in the train domain $\mathcal{X}^{train}$ to any random permutation of images in the test domain (a form of overfitting). This is because the objective is highly unconstrained. As a result, we use a cycle-consistency loss (Zhu et al., 2017; Hoffman et al., 2018), which uses a new mapping $\mathbf{G}' : \mathcal{X}^{test} \to \mathcal{X}^{train}$ which will take as input the translated instance $\mathbf{G}(\mathbf{x})$ and try to invert this function to get back the original instance, i.e., $\mathbf{x} \to \mathbf{G}(\mathbf{x}) \to \mathbf{G}'(\mathbf{G}(\mathbf{x})) \approx \mathbf{x}$. Using an L1-loss the task-cycle-consistency loss is given as:

$$\mathcal{L}_{cycle} = \mathbb{E}_{\mathbf{x}^{train} \sim \mathcal{D}_{train}}[||\mathbf{G}'(\mathbf{G}(\mathbf{x})) - \mathbf{x}||_1] \tag{6}$$

Combining the objectives from equation 5 and equation 6, we get the domain adaptation objective as $\mathcal{L}_{da} = \mathcal{L}_{GAN} + \mathcal{L}cycle$.

#### 3.3.2 ADDITIONAL IMPROVEMENTS

The objective in equation 1 is the basic objective of our proposed framework. We also consider two advanced variants that help improve the performance of the domain adaptation. First, we consider an *identity* loss where we encourage $\mathbf{G}$ to behave like an identity mapping when it receives an instance from $\mathcal{X}^{test}$, thereby behaving as an identity function for a test task. We also introduce a reverse direction mapping to map instances from test tasks $\mathcal{X}^{test} \to \mathcal{X}^{train}$, and a corresponding cycle loss to reconstruct back the instance in $\mathcal{X}^{test}$. All these objectives get tied together to deliver an appropriate task-level domain adaptation for a few-shot learning task. These improvements follow from state of the art image-to-image translation techniques (Taigman et al., 2017; Zhu et al., 2017).

## 4 EXPERIMENTS

### 4.1 EXPERIMENTAL SETTING

The setting we follow is: we have meta-training data in the original domain, unlabeled data in the target domain which is used for domain adaptation, and the meta-test data, from which test tasks will be drawn. There is no overlap between the data used for domain adaptation, and the meta-test data. Being a new problem setting, there have not been any approaches in literature directly addressing this problem. In order to be comparable, we adapt some of the techniques in Meta-Learning and Domain Adaptation, to make them suitable for our setting. Specifically, we consider 3 state of the art domain adaptation baselines RevGrad (Ganin et al., 2016), ADDA (Tzeng et al., 2017), and CyCADA (Hoffman et al., 2018). These baselines are trained on the meta-train data to learn a multi-class classifier and the unlabeled target domain data is used for domain adaptation. During meta-testing, these models are used as feature extractors, and K-NN is performed for prediction. We consider two meta-learning baselines MAML (Finn et al., 2017) and Prototypical Networks (Snell et al., 2017), which are trained on meta-train data; however these approaches do not consider the domain-shift issues. We construct a baseline that combines meta-learning with task-level domain shift. It is a combination of Prototypical Networks (Snell et al., 2017) with Gradient Reversal (Ganin et al., 2016), which we call *Meta-RevGrad*. Meta-RevGrad jointly optimizes PN-loss and a domain-confusion loss at the feature level where the embedded features of training tasks are made to appear like embedded features of test tasks resulting in the objective: $\lambda \mathcal{L}_{fs} + (1 - \lambda)\mathcal{L}_{RevGrad}$. Readers are refered to Ganin et al. (2016) for greater detail on $\mathcal{L}_{RevGrad}$. Here $\lambda \in (0, 1)$ is a trade-off parameter between few-shot performance and domain adaptation. We try several values of $\lambda = 0.9, 0.8, 0.7, 0.6, 0.5$ and report the best result. For our proposed method for Meta Learning with Domain Adaptation: MLDA, we consider three variants: the basic version *MLDA* based on equation 1; *MLDA+idt*, which considers the previous objective and an identity loss (see Section 3.3.2); and *MLDA+idt+revMap* which adds an additional component of (reverse) mapping testing tasks to train tasks (see Section 3.3.2).

Most of our code was implemented in PyTorch (Paszke et al., 2017) (for both baselines and proposed method). We follow the same model size and parameter setting for our models as the ones used in prior work for Prototypical Networks (Snell et al., 2017) and CycleGAN (Zhu et al., 2017). Jointly optimizing the objective in equation 1 can be very noisy (oscillating) and slow. To ease the implementation, we follow a two-step procedure for the optimization. We first optimize the objective with respect to all parameters except $\hat{\mathbf{F}}$. Then, all of these parameters are frozen, and $\hat{\mathbf{F}}$ is learned. Another issue with the GAN-based training is that the task generator lacks randomness, and always maps the same input task to the same output task (which can limit the meta-learning efficacy if the test-task domain is very diverse). To address this, during the GAN training we store intermediate models (e.g. a model saved after every epoch) and generate tasks using each of these models. This is similar to Snapshot Ensembles (Huang et al., 2017), where multiple models under one training cycle to increase robustness. We provide greater detail on the implementations in the appendix.

### 4.2 RESULTS ON CHARACTER RECOGNITION

We use Omniglot dataset (popularly used for benchmarking few-shot classification). The dataset comprises over 1,600 hand written characters, with 20 instances each. The dataset was further expanded by applying rotations. Inspired by a popular domain adaptation benchmark: MNIST to MNIST-M (Ganin et al., 2016), we design a new benchmark, suited for the few-shot learning under domain-shift: Omniglot to Omniglot-M. Omniglot-M is constructed in the same manner as MNIST-M, i.e., by randomly blending different Omniglot characters with different color background from BSDS500 (Arbelaez et al., 2011). We follow the same meta-train and meta-test split as previous approaches, (but meta-train and meta-test are in different domains). Unlabeled data from validation split (mutually exclusive with meta-test) is used for domain adaptation. We consider Omniglot to Omniglot-M, and Omniglot-M to Omniglot. We evaluate 1-shot, 5-class and 5-shot, 5-class tasks.

The results can be seen in Table 2. We can see that the basic domain adaptation and meta-learning approaches are not able to get a very good performance, as domain adaptation approaches are not suitable for few-shot learning, and meta-learning methods do not account for domain-shift. Meta-RevGrad is able to occasionally offer some improvement over the basic techniques, but is outperformed by our proposed MLDA. In general, MLDA can outperform all the baselines by a big margin. This can be observed in the case of both Omniglot → Omniglot-M and Omniglot-M →

Omniglot. Similar performance trends are observed for both 1-shot and 5-shot tasks. Within the variants of MLDA, we see that identity loss, and the reverse mappings are able to offer substantial boost to the overall performance, indicating better quality task-level domain adaptation.

Table 2: Few-Shot Classification Result via Meta Learning with Domain Adaptation on held-out Omniglot characters drawn from a different domain. Best performance is in bold.

| Method | Omniglot → Omniglot-M | | Omniglot-M → Omniglot | |
|---|---|---|---|---|
| | 1-shot, 5-way | 5-shot, 5-way | 1-shot, 5-way | 5-shot, 5-way |
| **RevGrad** (Ganin et al., 2016) | 26.68% | 29.15% | 69.89% | 85.29% |
| **ADDA** (Tzeng et al., 2017) | 27.18% | 34.45% | 69.10% | 86.15% |
| **CyCADA** (Hoffman et al., 2018) | 28.97% | 40.30% | 83.08% | 95.18% |
| **MAML** (Finn et al., 2017) | 26.22% | 30.46% | 74.14% | 83.41% |
| **PN** (Snell et al., 2017) | 27.66% | 34.46% | 74.23% | 88.92% |
| **Meta-RevGrad** | 25.97% | 35.51% | 71.70% | 85.50% |
| **MLDA** (ours) | 52.17% | 73.32% | 74.35% | 90.31% |
| **MLDA+idt** (ours) | 55.14% | 77.79% | 92.15% | **98.75%** |
| **MLDA+idt+revMap** (ours) | **58.35%** | **80.01%** | **94.91%** | 98.40% |

### 4.3 RESULTS ON OFFICE-HOME DATASET

We conducted similar experiments using Office-Home Dataset (Venkateswara et al., 2017), in particular data from two domains: Clipart and Product. There are a total of 65 classes, and we randomly split them into 3 sets, labelled data for meta-train (25 classes), unlabeled data for domain adaptation (20 classes), and the meta-test data (20 classes). All images were resized to 84x84x3, and all models were trained from scratch (pretrained models were not used). We consider Clipart to Product, and Product to Clipart. We evaluate 1-shot, 5-class and 5-shot, 5-class tasks.

The results can be seen in Table 3. The observations here are similar in trend to those observed for the character recognition experiments. The basic meta-learning approaches are quite poor (even though better than random). Meta-RevGrad can offer some improvement, and MLDA gives an even better performance. The performance trend is fairly consistent for both 1-shot and 5-shot tasks.

Table 3: Few-Shot Classification Result via Meta Learning with Domain Adaptation on Clipart and Product Domains from Office-Home dataset. Best performance is in bold.

| Method | Clipart → Product | | Product → Clipart | |
|---|---|---|---|---|
| | 1-shot, 5-way | 5-shot, 5-way | 1-shot, 5-way | 5-shot, 5-way |
| **RevGrad** (Ganin et al., 2016) | 25.42% | 43.11% | 27.05% | 36.69% |
| **ADDA** (Tzeng et al., 2017) | 31.99% | 42.57% | 27.63% | 31.17% |
| **CyCADA** (Hoffman et al., 2018) | 30.48% | 51.08% | 29.20% | 44.04% |
| **MAML** (Finn et al., 2017) | 35.75% | 51.12% | 32.15% | 44.14% |
| **PN** (Snell et al., 2017) | 36.25% | 52.84% | 32.62% | 44.48% |
| **Meta-RevGrad** | 37.36% | 52.84% | 33.59% | 46.61% |
| **MLDA** (ours) | 39.87% | 54.44% | 34.30% | 46.89% |
| **MLDA+idt** (ours) | **41.26%** | 53.31% | 34.50% | 47.82% |
| **MLDA+idt+revMap** (ours) | 39.19% | **55.93%** | **34.86%** | **47.96%** |

## 5 CONCLUSION

In this paper we investigated a novel problem setting: Meta-Learning for few-shot learning under task-level domain shift. Existing meta learning paradigm for few-shot learning was designed under the assumption that both training tasks and test tasks were drawn from the same distribution. This may not be the case for real world applications, where researchers may not find ample labeled data to simulate training tasks to be drawn from the same distribution as their test tasks. To alleviate this, we propose a meta learning with domain adaptation paradigm, which performs meta-learning by incorporating few-shot learning and task-level domain adaptation unified into a single meta-learner. We instantiate our few-shot model with Prototypical Networks and adopt an adversarial approach for task level domain adaptation. We conduct several experiments to validate the proposed ideas.

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

# 6 APPENDIX: DATASET CONSTRUCTION

## 6.1 OMNIGLOT ↔ OMNIGLOT-M

Here we show the details of the original Omniglot dataset and the statistical details, and some examples of how the characters look in a different domain. The meta-train, meta-test, and domain adaptation split of classes we used are based on the same split used in prior work. There is no overlap of classes or instances among the three sets, i.e., they are all mutually exclusive both at instance and class-level.

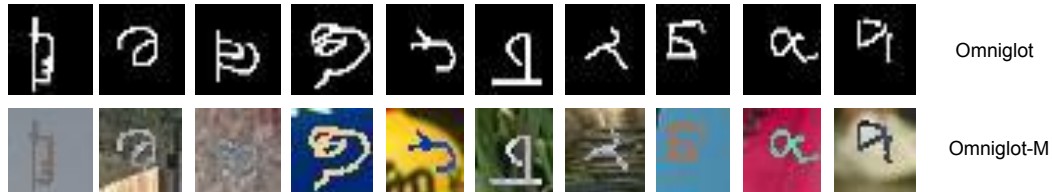

Figure 3: Example images of Omniglot and Omniglot-M

Table 4: Details on the Omniglot and Omniglot-M dataset used for benchmarking Meta Learning with Domain Adaptation.

| Domain | Split | #Classes | #Images |
|--------|-------|----------|---------|
| Omniglot | Meta-train | 4,112 | 82,240 |
| | Unlabeled Data in Target Domain | 1,692 | 33,840 |
| | Meta-test | 688 | 13,760 |
| Omniglotm | Meta-train | 4,112 | 82,240 |
| | Unlabeled Data in Target Domain | 1,692 | 33,840 |
| | Meta-test | 688 | 13,760 |

## 6.2 CLIPART ↔ PRODUCT

Here we show details of Office-Home dataset and some examples of how classes look in a different domains (Clipart and Product). The meta-train, meta-test, and domain adaptation split of classes used by us is shown in Tables 5 and 6.

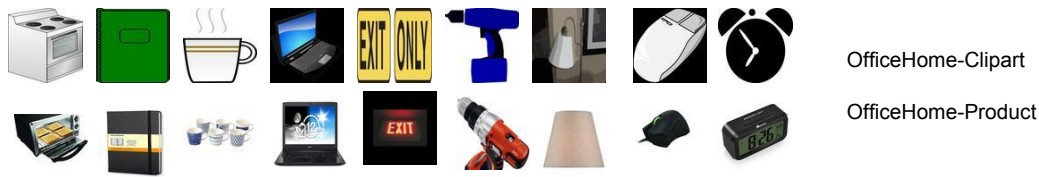

Figure 4: Details on the Clipart and Product domains used from the Office-Home Dataset.

# 7 APPENDIX: MODEL CONFIGURATION AND TRAINING

For MLDA, we followed training procedures adopted similar to Zhu et al. (2017) and Snell et al. (2017). Specifically, for CycleGAN, we changed the parameters related to image dimensions (scaling and cropping pre-processing) to keep the generated image size fixed to the original size i.e. $28 \times 28$ for Omniglot/Omniglot-M and $84 \times 84$ for OfficeHome Clipart/Product. The generative networks are the same as the original work (Zhu et al., 2017), each including two stride-2 convolutions with residual blocks, and two fractionally-strided convolutions with stride $\frac{1}{2}$. For all experiments, we used 6 blocks to generate images. We initialized the learning rate to 0.0002 and kept this learning rate for training till 100 epochs. The model after each epoch was used to translate source task to

Table 5: Details on the OfficeHome-Clipart and OfficeHome-Product dataset used for benchmarking Meta-Domain Adapatation.

| Domain | Split | #Classes | #Images |
|---|---|---|---|
| Product | Meta-train | 25 | 1,701 |
| | Unlabeled Data in Target Domain | 20 | 1,410 |
| | Meta-test | 20 | 1,328 |
| Clipart | Meta-train | 25 | 1,746 |
| | Unlabeled Data in Target Domain | 20 | 1,250 |
| | Meta-test | 20 | 1,369 |

Table 6: Details split of classes used for training and testing.

| Train | | | Test | | Domain Adaptation | |
|---|---|---|---|---|---|---|
| Calculator | Spoon | Radio | Shelf | Batteries | Printer | Paper_Clip |
| Flowers | Drill | Pencil | Scissors | Soda | Glasses | Bike |
| Knives | Table | Exit_Sign | Calendar | Sink | Marker | Sneakers |
| Laptop | Folder | Couch | Chair | Webcam | Mouse | Notebook |
| Pan | ToothBrush | Candles | Bed | Eraser | Postit_Notes | Toys |
| Clipboards | Fan | | Screwdriver | Hammer | Lamp_Shade | Helmet |
| Curtains | Pen | | Flipflops | Ruler | Speaker | Alarm_Clock |
| Telephone | Mug | | Mop | Bottle | TV | Refrigerator |
| Oven | Backpack | | Trash_Can | Monitor | Kettle | Desk_Lamp |
| Computer | Keyboard | | Fork | Bucket | File_Cabinet | Push_Pin |

target task. Weights were initialized with Gaussian distribution with mean 0 and standard deviation 0.02. We use the Adam solver with a batch size of 1. We also modified the loss function for diffent settings of MLDA. Specifically, for *MLDA*, we removed the losses related to target → source (B → A) mapping and set $\lambda_{idt} = 0$. For *MLDA+idt*, we set $\lambda_{idt} = 0.1$. For *MLDA+idt+revMap*, we kept the loss function the same as the original CycleGAN.

For Prototypical Networks, we followed the best hyperparameter settings in Snell et al. (2017). We used the same embedding architecture in the original work, including four convoluational blocks, each of which comprises a 64-filter $3 \times 3$ convolution, batch normalization layer, ReLU activation, and $2 \times 2$ max-pooling layer. This results in 64-dimensional output space for Omniglot/Omniglotm and 1600-dimensional output space for HomeOffice Clipart/Product. For Omniglot/Omniglotm experiments, the learning rate was set to 0.001 and reduced by half every 2K iterations, starting from iteration 2K. The network is trained for a total of 20K iterations. For OfficeHome Product/Clipart experiments, we initialized the learning rate to 0.001 and decayed the learning rate by half every 25K iterations, starting from iteration 25K. The model is trained up to 100K iterations. We also use Adam solver to optimize the networks. Following Snell et al. (2017), we chose squared Euclidean distance to perform kNN classification as this metric showed superior performance in prior work.

For all the baselines, we reused the official code and ran them with default hyperparameters. We only modified parameters to make the models compatible with the image resolution and number of classes in Omniglot/Omniglot-M and Product/Clipart datasets. In all experiments, we set $N_c$ classes and $N_S$ support points per class identical at training and test-time. We also fixed 15 query points per class per episode in all experiments. We computed the classification accuracy by averaging over 600 randomly generated episodes from the Meta-test set.

