# OpenReview forum: "Meta-Learning with Domain Adaptation for Few-Shot Learning under Domain Shift"
_ICLR.cc/2019/Conference_

### Official Review · AnonReviewer1 · 2018-11-02
**Interesting setting. Method seems to work, though is not very principled. Questions about reproducibility**

**Rating:** 6
**Confidence:** 3

**Review:**

The authors consider the few-shot / meta-learning scenario in which the test set of interest is drawn from a different distribution from the training set. This scenario is well-motivated by the "researcher example" given throughout the paper. The authors assume access to a large unlabelled set in test (target) domain, and a large labelled (few-shot) set in the source domain. Thus, the paper is concerned with unsupervised version of the meta-learning problem under domain shift (i.e., a large amount of data unlabelled are available from the target domain).

The key idea is to learn a mapping from the source domain to the target domain. This mapping is learned jointly with the meta-learner, who performs the meta-learning in the target domain, on examples from the labelled domain. In practice however, it appears from the experimental section that the domain mapping is learned offline, and then frozen for the meta-learning phase.  Thus, at test time, given examples from the target domain, the meta-learner can perform few-shot learning.

Pros:
- The paper addresses an important scenario which has not been addressed to this point: namely, meta-learning without the assumption that the train and test sets are drawn from the same domain/distribution.
- The authors propose a novel task and experimental framework for considering their method, and show (somewhat unsurprisingly) that their method outperforms standard meta-learning methods that do not properly account for domain shift.
- The paper reads well and is easy to follow.

Cons:
- My main concern is reproducibility: the authors employ a number of large architectures, complex loss functions, and regularizers / "additional improvements". Further, there a number of experimental details that need to be further elaborated upon. e.g., architectures and hyper-parameters used, and training procedures (I encourage the authors to utilize the appendices for this). It is unclear to me how difficult/easy these results would be to reproduce. Do the authors intend to release code for their implementations and experiments?
- Some assumptions are not explicitly stated. In particular, it is unclear what the assumption on the size of the unlabelled test set is. This is also lacking from the description of the experimental protocol, which does not address the data-splits (how many classes were used for each) and size of the unlabelled test set.
- While the method is presented as jointly learning all the components, in the experimental section it is stated that the embedding network (the meta-learner) and the GAN-based domain adaptation are done separately. Can the authors comment on this further? Is this different from first learning a image translation mapping (using the unlabelled data in the target domain), and then applying existing meta-learning models/algorithms to the labelled data in the target domain?
- The overall method seems to be not very principled, and requires a lot of "tweaks and tunes", with additional losses and regularizers, to work.

Overall, the paper proposes a method combining a number of existing useful works (prototypical networks for meta-learning and image-to-image translation for domain adaptation) to tackle an important problem setting that is not currently addressed in existing meta-learning research. Further, it establishes a useful experimental benchmark for this task, and provides what appear to be reasonable results (though this is somewhat difficult to judge due to the lack of baseline approaches). Hopefully, such a benchmark will inspire more researchers to explore this setting, and perhaps propose simpler, more principled approaches to perform this task. It is my impression that, if the authors elaborate on the experimental protocol and implementation details, this paper would be a good fit for the venue.

---

> ### Author Response · Authors · 2018-11-26
> **Thanks for the useful comments, we have made suitable updates**
>
> Overall
> We thank you for your kind comments, in acknowledging that the work is well motivated, and the problem is an important one, currently not studied under the meta-learning paradigm. We have made substantial improvements based on your suggestions, and other reviewers’ comments, and hopefully we are able to address most of your concerns.
>
> Concern 1: Reproducibility
> Our code is built on top of existing code (Prototypical Networks and Image-to-Image Translation from CycleGAN). Thus, we adopt the same hyperparameters and architectures as the prior work, and as a result our work is fairly easy to reproduce. We will of course release the code. As suggested, we have utilized the appendices to give detailed information about the experimental setup.
>
> Concern 2: Size of unlabelled test set, data-split information
> We did provide some details on the first version (in the appendix). In light of the reviews, in the revised version, we have expanded the appendix to give more details on the experimental protocol.
>
> Concern 3: Jointly learning vs Freezing GAN
> Training in joint manner can be very tricky, and may often cause stability issues. You are right in your suggestion, that it is similar to first styling, and then applying meta-learning. Having said that, this is a common strategy in several state of the art domain adaptation techniques, where the GAN-based domain adaptation and task-specific classifier are trained in multiple steps. For example, see training protocol in [1,2,3, etc.].
>
>
> [1] Judy Hoffman, Eric Tzeng, Taesung Park, Jun-Yan Zhu, Phillip Isola, Kate Saenko, Alyosha Efros, and Trevor Darrell. Cycada: Cycle-consistent adversarial domain adaptation. In ICML, 2018
> [2] Eric Tzeng, Judy Hoffman, Kate Saenko, and Trevor Darrell. Adversarial discriminative domain adaptation. In Computer Vision and Pattern Recognition (CVPR), volume 1, pp. 4, 2017.
> [3] Bousmalis, K., Silberman, N., Dohan, D., Erhan, D., & Krishnan, D. Unsupervised pixel-level domain adaptation with generative adversarial networks. In The IEEE Conference on Computer Vision and Pattern Recognition (CVPR), 2017

---

### Official Review · AnonReviewer2 · 2018-11-02
**This is not "meta domain adaptation" but "few-shot learning +domain adaptation"**

**Rating:** 5
**Confidence:** 3

**Review:**

This paper proposes to combine unsupervised adversarial domain adaptation with prototypical networks and finds that the proposed model performs well on few-shot learning task with domain shift, much better than other few-shot learning baselines that do not consider. Specifically it tests on Omniglot with natural image background and cliparts to real images.

It is true that current meta-learning approaches do not address the problem of domain shift, and as a result, the testing domain has to be the same with the training domain. However, this paper rather than proposing solution address the meta-learning problem, albeit the title “meta domain adaptation”, only brings few-shot learning to domain adaptation. Here’s why:

In order for a meta-learning model to be called “meta domain adaptation,” the type of adaptation cannot be seen during training, and the goal is to test on adaptation that the model has not seen before. Indeed, each task in meta domain adaptation should be seen as a pair of source task and target task.

The problem with the current model is that during training, it is trained to target at one specific type of test domain--the generator network G aims to generated images that align with the unsupervised  test domain X_test. Thus, the trained model will also only be able to handle one test domain, not much different than regular meta-learning models.

In short, the meta-learning part stays in the regular few-shot learning module (which is implemented as a prototypical network), and has nothing related to domain adaptation. Therefore, the paper cannot be qualified for ``meta domain adaptation’’ and has very limited novelty in terms of its contribution to meta-learning; however, the combination of domain adaptation and few-shot learning is fair. For the rest of my review, I will treat the paper as “few-shot learning with domain adaptation” for more appropriate analysis.

For the experiments, there seems to have a great win of the proposed algorithm against the baselines. However, I think since this is few-shot learning with domain adaptation, there is no domain adaptation baselines being mentioned in comparison. Specifically, what if the few-shot learning component is removed, and the network is trained with standard domain adaptation. Then use the same network to extract the features and then using the nearest neighbor to retrieve the classes. Also it seems that the regular batch normalization could be very sensitive to domain shifts, and it would be good if the authors can test other normalization schemes such as layer/group normalization as baselines.

Another concern is that the evaluation of domain adaptation does not have much varieties. Only two domains shifts are evaluated in the paper, specifically Omniglot + BSD500 and Office-Home. BSD 500 only contains 500 images, and it would be good if more diverse set of images are considered. Other domain transfer settings such as synthetic rendered vs. real (e.g. visDA challenge) could have been considered.

In conclusion, the paper presents a interesting combination of ProtoNet + Adversarial DA + Cycle consistency. However, unlike as advertised, the paper does not address the domain shift issue in meta-learning, and the experiments lack thorough evaluation as the paper considers itself as a meta-learning paper and only compares to other meta-learning approaches without much comparison to domain adaptation papers. Therefore, I recommend reject.

---
Note: after reading the comments updated by authors, I remain my opinions: even though exact meta-testing data is unseen during training, the domain is seen during training, and therefore it cannot be qualified for being "meta domain adaptation".

===
After rebuttal:

I would like to thank the authors for the response and updating the draft. They have addressed 1) the title issue and 2) adding domain adaptation baselines. Considering these improvements, I would like to raise the score to 5, since the setting of combining few-shot learning and domain adaptation is interesting and the proposed model outperforms the baselines.

However, my criticisms remain that the paper is a simple combination of cycle GAN and prototypical networks, and lacks new insights/novelty. The experiments use fairly small datasets, where the performance can be largely influenced by how good the feature extractor backbone is (e.g. training on more data and using deeper architecture would warrant better performance, and thus may change the conclusion).

---

> ### Author Response · Authors · 2018-11-26
> **We agree with your concerns about possible misinterpretation, but we think it is fixable; Thanks for your suggestions on DA baselines - we have incorporated them in the revision**
>
> Overall:
> We thank you for your valuable suggestions in helping us avoid potential inefficiencies in our work, and suggesting ways to avoid misunderstandings. We have incorporated your comments to significantly improve our work, and hope our revised draft is able to convince you towards a favorable outcome.
>
> Concern 1: Concerns with title “Meta Domain Adaptation”
> “…unlike as advertised, the paper does not address … “
>
> It appears that our choice of the title may have resulted in the reviewer qualifying our paper as a form of false advertising. We acknowledge this problem and agree with you about a possible misinterpretation. However, we feel this maybe a bit harsh, as in the technical content of the paper (Abstract, Introduction, etc.), we have been very clear about the motivation and the problem setting, and do not think we did any form of false advertising.
>
> We also do think that the problem setting we have proposed is an important problem that deserves attention, and has not been studied in the meta-learning paradigm. We are glad that you also agree that setting makes sense (“... the combination … is fair”). Overall, we think that we have made an important contribution to Meta-Learning literature, by identifying its limitation for few-shot learning under domain shift, and proposed a solution to tackle this problem.
>
> We have tried to revise the draft with appropriate renaming of the method to avoid potential misunderstandings. In fact, we have mostly changed the name from “Meta Domain Adaptation” to “Meta Learning with Domain Adaptation”, and the rest of the paper is almost identical, which we believe addresses the concerns of false advertising.
>
>
> Concern 2: Experiments
> Domain Adaptation Baselines + Other datasets
>
> Being a new problem setting, designing appropriate baselines can be challenging. We considered the traditional meta-learning for few-shot learning approaches, and combined meta-learning with a popular domain adaptation baseline.
>
> We are grateful for your suggestions on the domain adaptation baselines, and fully agree that it is reasonable. It is something we should have done on our own. Accordingly, based on your suggestions, and suggestions from other reviewers, we have tried to expand the baselines substantially (specifically, we include three state of the art Domain Adaptation methods as baselines – RevGrad [1], ADDA [2] and CyCADA [3]), and our proposed methods outperform them.
>
> For the other dataset suggested (VisDA), for synthetic-real adaptation, it is difficult to match the training paradigm of meta-learning. Typically, we desire several classes for meta-train, and several classes for meta-test, so that a variety of (e.g.) 5-way tasks can be crawn. With just 12 classes, the dataset is not very suitable for such settings.
>
> [1] Ganin, Yaroslav, et al. "Domain-adversarial training of neural networks." The Journal of Machine Learning Research 17.1 (2016): 2096-2030
> [2] Tzeng, Eric, et al. "Adversarial discriminative domain adaptation." Computer Vision and Pattern Recognition (CVPR). Vol. 1. No. 2. 2017
> [3] Hoffman, J., Tzeng, E., Park, T., Zhu, J. Y., Isola, P., Saenko, K. & Darrell, T. Cycada: Cycle-consistent adversarial domain adaptation. ICML 2018

---

> ### Author Response · Authors · 2018-12-02
> **Further discussion - response to post-rebuttal updated review**
>
> We thank you for considering our rebuttal and updating the score. We are grateful for your time and advice, and would appreciate if we could further extend the discussion.
>
>
> We appreciate the concern in the updated comments, but would like to point out that the novelty in our work should be viewed from two angles: the need to study this problem (i.e., the problem setting), and the proposed solution. We have identified a novel problem setting, which is closer to the real world setting, than what has been studied so far under the meta-learning paradigm. Existing solutions are not effective in this setting, restricting their use in the real world. Addressing this setting in our framework gives us a direction to improve the practical utility of meta-learning solutions for few-shot learning. Specifically, we identify that the principle of image-to-image translation is very suitable for this setting, and apply those concepts to boost the performance of few-shot learning under domain shift. As a combination of problem setting and proposed solution, we do believe we have addressed an important problem, and made a novel contribution.
>
> As regards the experiments: “fairly small datasets … feature extractor backbone”
>
> Most domain adaptation experiments use MNIST, USPS, SVHN, which are comparable in size to our Omniglot experiments. The other popular benchmark is using the Office-dataset, which also we have used (although a more recent version of a similar dataset, i.e., office-home – more suitable for meta-learning evaluation, as it has larger number of classes). See for example some of the recent domain adaptation papers [1, 2, 3].
>
> While a feature extractor backbone network may have some influence, we would like to highlight three points. First, when networks are trained in one domain, and evaluated in another, regardless of the backbone network, it is the domain-shift that dominates the performance. For example, no matter how large the network is, if it is trained to recognize black and white digits, it will still struggle to recognize colored digits. Second, any benefit of a larger backbone network will likely also enhance the performance of our model. Third, we just wanted to clarify (if there was a misunderstanding), unlike domain adaptation papers, we do not use a pretrained network – we train the full network from scratch (following traditional meta-training settings).
>
> [1] Ganin, Yaroslav, et al. "Domain-adversarial training of neural networks." The Journal of Machine Learning Research 17.1 (2016): 2096-2030
> [2] Tzeng, Eric, et al. "Adversarial discriminative domain adaptation." Computer Vision and Pattern Recognition (CVPR). Vol. 1. No. 2. 2017
> [3] Hoffman, J., Tzeng, E., Park, T., Zhu, J. Y., Isola, P., Saenko, K. & Darrell, T. Cycada: Cycle-consistent adversarial domain adaptation. ICML 2018

---

### Official Review · AnonReviewer3 · 2018-11-03

**Rating:** 6
**Confidence:** 3

**Review:**

The authors proposed meta domain adaptation to address domain shift scenario in meta learning setup. The proposed model combines few shot meta-learning with the adversarial domain adaptation to demonstrate performance improvements in several experiments.

Pros:
1. A new few shot learning with domain shift problem is studied in the paper.
2. A new model combining prototypical network with GAN and cycle-consistency loss for addressing meta-learning domain shift scenario. The experimental improvements on omniglot seem quite substantial.

Cons:
1. Can you clarify why the proposed approach is better than the Meta-RevGrad baseline? It seems that both are using meta-learning with domain adaptation technique. What happen for Meta-RevGrad + idt or Meta-RevGrad + revMap?  I feel the baseline in domain adaptation area is a bit limited.
2. How is the performance of a simpler baseline such as combining a subset of new domain as training set to train MAML or PN (probably in 5-shot, 5-class case)?
3. It seems the domain shift in the paper is less dramatic. i.e., omniglot <-> omniglot-M. I wonder whether the proposed approach can still work in large domain shift such as omniglot to fashion-mnist etc.
4. The novelty of the model is relatively limited as it is a combination of previous techniques on a new problem.

Minor:
1. Where is L_da in Figure 2? In Figure 2, what’s the unlabelled data from which testing tasks are drawn? Is it from meta-test data training set?
2. In the caption of figure 2, there should be a space after `":".

---

> ### Author Response · Authors · 2018-11-26
> **Thanks for the suggestions, we have tried to improve the draft accordingly**
>
> Overall:
> We thank you for your time and appreciating the strengths of our work. Based on your guidance and suggestions, we have tried to do additional experiments to improve the quality of our work.
>
> Concern 1: Comparison to Meta-RevGrad
> Meta-RevGrad tries to achieve feature invariance at the embedding level. Achieving such feature invariance for high-dimensional feature mapping can be a very weak constraint [1], causing limitation in performance. Recently, generative approaches, following image-to-image translation have been shown achieve better domain adaptation, as this constrains the feature embedding to generate the data in a new domain. Being a non-GAN based approach, concepts such idt (encouraging the styling network to behave as an identity when given a target domain instance as input) and revMap (constructing source instance back from generated target instance) are not applicable in this scenario, as no instances or images are being generated from a feature embedding.
>
> Concern 1, 2, 3: Experiments
> Thank you for these suggestions. Taking your comments and comments from other reviewers into account, we have made improvements to the experimental section. Specifically:
>
> “Domain Adaptation Baselines”,
> We have now added additional domain adaptation baselines, designed in the setting suggested by Reviewer 2.
>
> “Simple baseline – combining a subset of a new domain as training set”
> We see the merit of this baseline, but there are several challenges in executing this. Designing it in a fair way is tricky. Using some labelled data in target domain maybe unfair, as we are not allowed to see meta-test data. Moreover, this is likely to not work, as the meta-train data would be too large, and would dominate, and we do not have a clear way to set the weights.
>
> “Dramatic Domain Shift, Omniglot to Fashion-MNIST”
> This could be an interesting setting, but we don’t think this will work very well, as the tasks are themselves completely different. We would not expect a character recognition model to transfer to a object recognition task, as the visual features are very different.
>
>
> Minor:
> Thanks for this; we have updated the draft to make the presentation clearer. Unlabelled data refers to only the domain of the meta-test data, but the meta-test data is never used in meta-training.
> L_da is essentially the sum of L_gan and L_cycle.
>
>
> [1] Shu, R., Bui, H.H., Narui, H. and Ermon, S. A DIRT-T Approach to Unsupervised Domain Adaptation. ICLR 2018

---

### Author Response · Authors · 2018-10-25
**Response to Blog Post Review of our paper**

Recently we happened to come across a public review of our ICLR submission on a personal blog post: https://zhuanlan.zhihu.com/p/46340382

In the spirit of Open Review, even though these comments were not posted on the official website, we believe we should take into account all suggestions we received to improve our research. While we do not fully agree with the post, we are grateful for some insightful comments that we think may contribute towards discussions, improving the quality of our work, and its interpretation.

Google Translate was used to translate the post to English. We have summarized a few main concerns, and accordingly provided our response and updates made to our work.

Concern 1 - Motivation
 “The paper repeatedly said that the data of the Meta-Testing Domain may be very small in reality…”
 “…However, this motivation is contrary to the actual paper method implementation, and the method does not limit the use of Meta-Testing Domain data. So the motivation of this article is a bit problematic…”

Response:
We would like to clarify, that we explicitly state that LABELED data in the meta-testing domain is scarce, meanwhile we think that it is reasonable to assume unlabelled data can be easily obtained in the target domain. In literature, it has been a commonly accepted setting that most domain adaptation methods use unlabelled data in the target domain to help overcome the domain shift. We operate under a similar setting, but aim to do few-shot learning. Machine Learning literature often assumes that unlabelled data (in any domain) is relatively inexpensive to obtain, and it can (and should be) leveraged to improve performance whenever possible.

The traditional meta-learning paradigm acquires few-shot learning ability from many training data, and effectively aims to perform transfer learning from this heavily labelled data onto a new task drawn from the test-task distribution. If in the real world, a researcher collects new data to perform few-shot classification for their own task, they are not able to do so because they still need to acquire a lot of labelled training data in the same domain to do meta-learning, and we argue this is an unrealistic demand and a critical limitation of current meta-learning paradigms. In this work, we propose a new direction towards making few-shot learning more practical and realistic, where labelled data that is available in one domain can be used to do meta-learning in a manner that it can be suitable for tasks in a new domain with limited amount of labelled data. We think that it is fairly reasonable to assume that unlabelled data in the target domain may be easily available.

Concern 2 – Experiments
“… motivation of this paper are contradictory, and it is unreasonable to use Meta-Testing data directly on Meta-Training.…”

Response

We thank the author of the post for pointing out one possible issue of our experiments, and we thus have tried to fix the issue in our experiments to improve the evaluation process. In particular, earlier in our experiments, the unlabelled data in target domain was used for domain adaptation, and test tasks were drawn from these unlabelled test data samples. Following the concern raised by the author of the post, we amend our evaluation protocol and create a setting, where the meta-test data from which tasks are drawn are completely unseen during the meta-training procedure. Instead, we use a third set of unlabelled samples where the data is drawn from the same domain as the test data, and these unlabelled data do not have any instances belonging to the same classes in the test tasks. Thus, meta-testing data will NEVER be used during the meta-train procedure. Following this setting, we performed the experiments again on Omniglot and Office-Home dataset, and we found that we can still obtain very similar results as before. The results as screenshots can be seen from this link: https://www.dropbox.com/s/71jx8davpsifcgj/iclr19_mda_results.JPG?dl=0

---

> ### Author Response · Authors · 2018-10-25
> **Response to Blog Post Review of our paper (2)**
>
> Concern 3 – Baselines
> “..,paper directly compares with MAML and Prototypical Network without domain adaptation…”
>
> Response
> This  concern is incorrect, since we have already implemented one baseline with domain adaption (developed by us), where Prototypical Networks are combined with Reverse Gradient (Ganin et. al. 2016), to achieve feature representation invariance between the domains. Nonetheless, whenever addressing a novel problem setting, there is always a room to improve experiments since baselines are often not readily available. We are planning to add some more baselines in the future work.
>
> Concern 4 – Alternate Setting
>  “… isn't the Train and Test Domain in Task different and more reasonable …”
>
> Response
> The problem setting proposed by the author of the post where Train and Test data of a task are in  a different domain makes sense, which has also attracted some attention very recently in literature under meta-learning paradigm (Yu et. al. 2018). However, their problem setting still make the assumption that the training tasks and test tasks are drawn from the same distribution, while we argue that this assumption may not always hold in practice, and we aim to address the challenging scenario where they are drawn from different distributions.

---

### Author Response · Authors · 2018-11-26
**Revision Log - for all reviewers**

Dear Reviewers,

We are extremely grateful for your comments, and we feel that they have really helped us improve the quality of our work. Below we give a summary of changes in the revised version, coded by the concerns raised by the reviewers.

Most edits are in the experimental section, to meet the 8-page constraint.

1.	Change of method name and title [AnonReviewer2]
We have changed the name of the method from “Meta Domain Adaptation” to “Meta-Learning with Domain Adaptation”. We have made appropriate changes to the content, such that possible misunderstanding from the title can be avoided. The changes are very minimal in the main draft.

2.	Experiments (1) – Changed Experimental Protocol to not see meta-testing data during meta-training [AnonReviewer3(minor)+ BlogReviewer]
We have revised the experimental protocol such that the meta-test data is never seen during training. Only unlabeled data from the target domain (from a different set of classes as the meta-test data) is observed during meta-training.

3.	Experiments (2) – Domain Adaptation Baselines [AnonReviewer2 + AnonReviewer3]
We have added three state of the art domain adaptation baselines in the experiments. Our proposed method is able to outperform these methods.

4.	Experiments (3) – Detailed Description of Experimental in Appendix [AnonReviewer1]
We have provided a detailed write-up on the experimental details in order to help reproducibility.


We thank you all for the extensive feedback, and hope that we were able to address your concerns.

Best regards,
ICLR 2019 Conference Paper368 Authors

---

### Meta-Review · Area_Chair1 · 2018-12-12
**Combination of prior work applied to a new problem statement -- more improvements needed**

**Confidence:** 4
**Recommendation:** Reject

**Metareview:**

This paper addresses the problem of few shot learning and then domain transfer. The proposed approach consists of combining a known few shot learning model, prototypical nets, together with image to image translation via CycleGAN for domain adaptation.  Thus the algorithmic novelty is minor and amounts to combining two techniques to address a different problem statement. In addition, as mentioned by Reviewer 2, though meta learning could be a solution to learn with few examples, the solution being used in this work is not meta learning and so should not be in the title to avoid confusion.

As this is a new problem statement the authors apply multiple existing works from few shot learning (and now adaptation) to their setting. The proposed approach does outperform prior work, however this is not surprising as the prior work was not designed for this task. Despite improvements during the rebuttal to address clarity the specific experimental setting is still unclear -- especially the setup of meta test data vs unsupervised da data.

This paper is borderline. However, since the main contribution consists of proposing a new problem statement and suggesting a combination of prior techniques as a first solution, the paper needs a more thorough ablation of other possible combination of techniques as well as a clearly defined experimental setup before it is ready for publication.